# Hydrogen solubility of stishovite provides insights into water transportation to deep Earth

**Mengdan Chen[1], Changxin Yin[1], Danling Chen[1], Long Tian[1], Liang Liu[1], and Lei Kang[1]**

[1]Department of Geology, Northwest University/State Key Laboratory of Continental Dynamics, Xi'an, 710069, China

*Correspondence to*: Lei Kang (kanglei@nwu.edu.cn); Liang Liu (liuliang@nwu.edu.cn)

**Abstract.** Water dissolved in nominally anhydrous minerals (NAMs) can be transported to deep regions of the Earth through subducting slabs, thereby significantly influencing the physicochemical properties of deep Earth materials and impacting dynamic processes in the deep Earth. Stishovite, a prominent mineral present in subducting slabs, remains stable at mantle pressures of 9-50 GPa and can incorporate various amounts of water ($H^+$, $OH^-$, and $H_2O$) in its crystal structure. Consequently,

stishovite can play a crucial role in transporting water into deep Earth through subducting slabs. This paper provides a comprehensive review of the research progress concerning water (hydrogen) solubility in stishovite. The key factors that govern water solubility in stishovite are summarized as temperature, pressure, water fugacity and aluminum content. Combined with published results on the dependence of water solubility on the aforementioned parameters, this paper proposes a new equation to describe the solubility of water in Al-bearing stishovite. Calculation results based on this equation suggest that

stishovite may effectively accommodate water released from processes such as hydrous mineral breakdown which could ultimately contribute to the presence of a water-rich transition zone.

## 1 Introduction

Water plays a crucial role not only in the origin and evolution of life but also in various Earth processes, including slab subduction, crust-mantle reaction and recycling. During slab subduction, water is transported from the Earth's surface to its

interior through subduction zones, and a significant amount of water returns to the surface mainly through magmatism, thus forming a large-scale water cycle (Fig. 1). In subducting slabs, hydrogen exists in melts, fluids, fluid inclusions, hydrous minerals (e.g., amphibole and mica) and nominally anhydrous minerals (NAMs) (Bell et al., 1995; Rossman, 1996; Xia et al., 1998; Johnson, 2006; Libowitzky, 2006; Litasov and Ohtani, 2007; Ni et al., 2017), in forms of both molecular water ($H_2O$) and structurally bound water ($H_3O^+$, $H^+$ and $OH^-$). Most hydrous minerals in subducting slabs decompose and release water

before reaching a depth of 300 km (Poli and Schmidt, 2002). Part of this water returns to the surface through magmatism, while the remainder is incorporated into NAMs in ultrahigh-pressure metamorphic rocks (Magni et al., 2014; Walter, 2021) and subsequently transported to deep mantle by subducting slabs (Ishii et al., 2022).

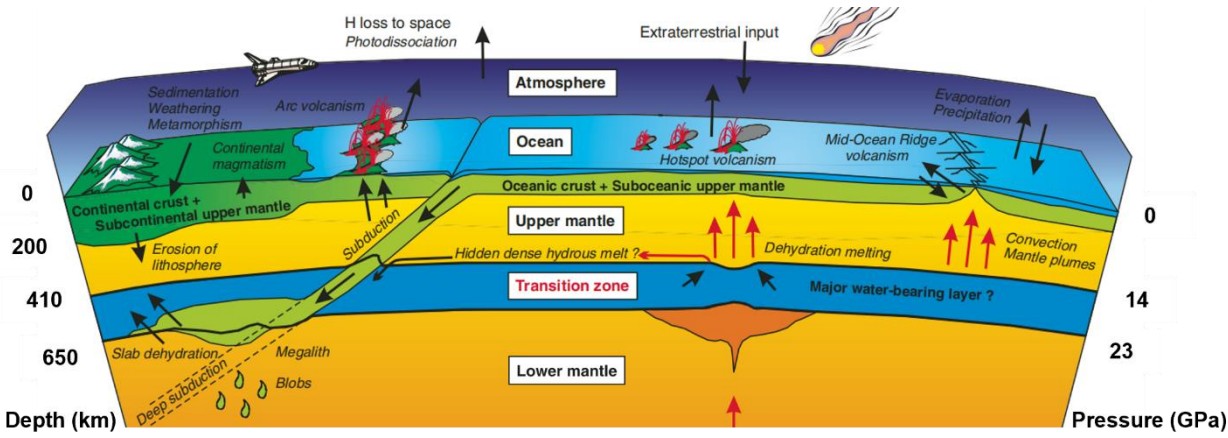

**Figure 1: Water transport, distribution, and cycling in the Earth (Litasov and Ohtani, 2007)**


    Numerous studies indicate that major constituent minerals in deep mantle, such as olivine, pyroxene, and garnet, as well as their high-pressure polymorphs, are NAMs (Litasov and Ohtani, 2007; Liu et al., 2016). Although the phase of $SiO_2$ does not typically appear in the mantle (Kaminsky, 2012), it is a significant constituent mineral in silica-riched slabs and can stably exist at upper to lower mantle depths through high-pressure polymorphic phase transitions (Fig. 2). Both experimental and

computational studies have demonstrated that stishovite can incorporate a certain amount of water. Given its prevalence as a major mineral in subducting slabs at mantle depths (>300 km), stishovite likely plays a significant role in water transportation to deep Earth (Spektor et al., 2011; Lin et al., 2022; Ishii et al., 2022). In this paper, we provide a systematic review of previous understandings and research progress regarding water solubility, water incorporation mechanisms, and factors influencing water solubility in stishovite. Considered stishovite in subducting slabs always contains amounts of Al in its crystal structure

(Ono, 1999; Lakshtanov et al., 2007a), this paper establishes an empirical model for water solubility in Al-stishovite based on published results, and discuss the significance of stishovite in transporting water from subducting slabs to the deep mantle and its implications for deep Earth dynamics. Finally, we highlight key unresolved scientific questions in the research on water solubility in stishovite.

## 2 Crystal structure and stability of stishovite

Phase of $SiO_2$ undergoes multiple high-pressure polymorphic transitions as pressure increases (Petersen et al., 2021) (Fig. 3). Under a subduction slab geothermal gradient, coesite is the dominant phase of $SiO_2$ at pressures ranging from 2.7 to 9 GPa. At approximately 10 GPa, coesite transforms into denser stishovite (Ono et al., 2017). The transition pressure from coesite to stishovite is dependent on temperature, as described by the equation P (GPa) = 4.7 + 0.0032 × T (K) (Ono et al., 2017). This phase transition is believed to contribute to the seismic discontinuity observed at around 300 km depth (Ono et al., 2017). As

pressure increases further to ~24 GPa, stishovite undergoes a transformation into the $CaCl_2$-type $SiO_2$ structure (commonly

referred to as post-stishovite) due to the incorporation of Al and H in stishovite (Ishii et al., 2022) . However, the transition pressure is still debated. Fischer et al. (2018) pointed out that the phase transition between stishovite and CaCl$_2$-type silica should occur at pressures of 68–78 GPa in the Earth, depending on the temperature in subducting slabs.

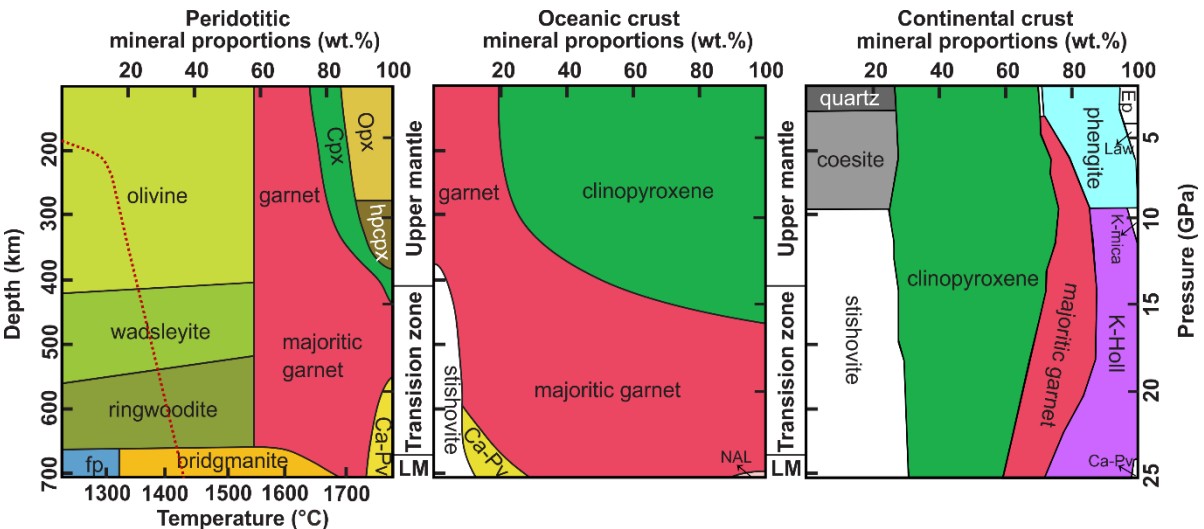

**Figure 2: Mineral assemblages in peridotite, subducted oceanic crust, and subducted continental crust. The red line indicates the mantle geotherm (Kaminsky, 2012). Modified after Smith et al., 2018 and Wu et al., 2009. (Fp—Ferropericlase; Opx—Orthopyroxene; Cpx—Clinopyroxene; Ca-pv—Ca-perovskite; NAL—Na-Al phase; Ep—Epidote; Law—lawsonite; K-Holl—K-Hollandite)**

Previous studies show that stishovite constitutes 10 vol. % and 20 vol. % of subducting mid-ocean ridge basalt (MORB) in the upper and lower mantle, respectively (Irifune et al., 1986; Ono et al., 2001). In subducted continental crust at upper mantle depths (<660 km), stishovite can reach approximately 20-25 vol. % (Irifune et al., 1994; Poli and Schmidt, 2002; Ishii et al., 2012). However, due to the exhumation of subducted slabs from ~300 km depth (termed the "depth of no return" in literature, e.g., Irifune et al., 1994; Liu et al., 2007; Wu et al., 2009) to the surface is extremely difficult, and stishovite is unstable and easily transforms to lower pressure SiO$_2$ polymorphs, naturally formed stishovite that can be observed is extremely rare. Previously, Yang et al. (2007) identified polycrystalline coesite as a potential pseudomorphic replacement of stishovite in Tibetan chromitites. The exsolution microstructure (Liu et al., 2007) and pseudomorphs after stishovite (Liu et al., 2018) were found in the South Altyn ultra-high pressure metamorphic belt in western China. Recently, Thomas et al. (2022) identified coesite and stishovite inclusions in the Waldheim granulite, and Gu et al. (2022) showed that coesite (former stishovite) was present within a natural super-deep diamond formed at the boundary between the transition zone and the lower mantle. In addition, other naturally occurring stishovite is found sporadically in meteorite impact craters (e.g., Chao et al., 1962) or meteorites (e.g., Holtstam et al., 2003). Therefore, our current understanding of stishovite mainly relies on high-temperature and high-pressure experiments and theoretical calculations (e.g., Litasov et al., 2007; Lin et al., 2022; Ishii et al., 2022).

Stishovite was first synthesized by Stishov and Popova (1961) at 20 GPa (equivalent to approximately 600 km depth in Earth's interior) and 1100 °C from α-quartz. It possesses a rutile-type structure with tetragonal symmetry (*P4₂/mnm*). The Si atoms are coordinated by six O atoms in octahedral arrangements (Pawley et al., 1993; Spektor et al., 2011; Lin et al., 2022). These SiO₆ octahedra align and form linear chains along the c-axis. This arrangement results in a highly dense packing of O atoms, with slightly elongated Si-O bonds compared to SiO₄ tetrahedra. Previous studies have demonstrated that the density

of stishovite is 46% higher than coesite and 60% higher than α-quartz, respectively (Keskar et al., 1991). Therefore, the formation of stishovite during slab subduction at depths exceeding 300 km can dramatically increase the density of subducting slabs (Lin et al., 2022; Ishii et al., 2022), and enhance their negative buoyancy for further subduction into the mantle transition zone (410-660 km) or even deeper regions.

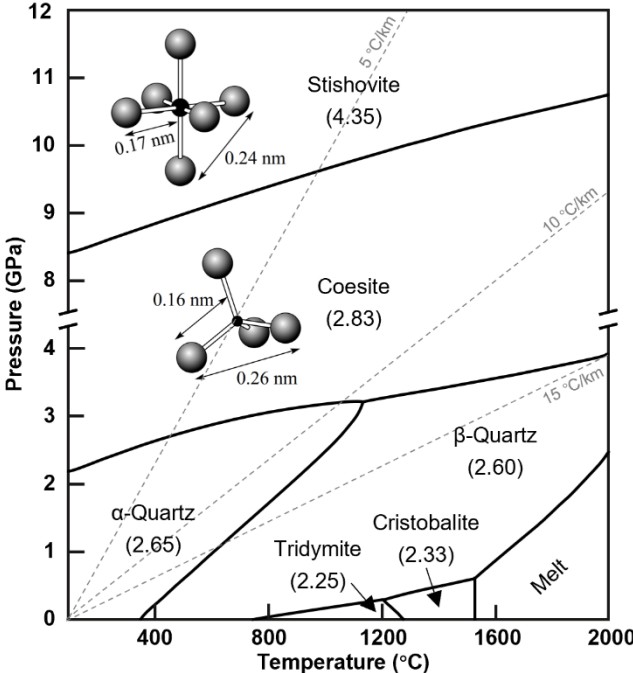


**Figure 3: SiO₂ phase diagram as recommended in the reference literature (density values are indicated in brackets) (Gutzow et al., 2014). The gray dashed line represents the subduction slab geothermal gradient (Zheng et al., 2016). The diagram also illustrates the SiO₄ tetrahedron and SiO₆ octahedron, along with their corresponding distances.**

# 3 Water solubility and incorporation mechanisms in stishovite

## 3.1 Water solubility

Research on water solubility in stishovite has primarily been conducted through high-pressure experiments (Chung and Kagi, 2002; Bromiley et al., 2006; Litasov et al., 2007; Spektor et al., 2011; Nisr et al., 2017; Lin et al., 2022; Ishii et al., 2022). Fourier transform infrared spectroscopy (FTIR) is the primary technique used to determine water content in experimental products, although results from different studies often show significant discrepancies, as detailed in Yan and Liu (2021). In 95 this paper, we compile the experimental conditions (temperature, pressure, initial water content) and results (water and $Al_2O_3$ contents in stishovite products) from previous studies (Table 1).

**Table 1** Water contents in stishovite from previous studies

| Temperature (°C) | Pressure (GPa) | Initial water content (wt. %) | Synthesis method | $Al_2O_3$ (wt. %) | Water content (wt. ppm) | Measurement method | References |
|---|---|---|---|---|---|---|---|
| 1200 | 10 | saturated | LVP | 0-1.51 | 7-82 | FTIR [a] | Pawley et al., 1993 |
| 1200-1500 | 15-21 | saturated | LVP | 0 | 2.5-72 | FTIR [b] | Bolfan-Casanova et al., 2000 |
| 1200-1400 | 10-15 | saturated | LVP | 0.612-1.341 | 210-759 | FTIR [b] | Chung and Kagi, 2002 |
| 1740-3700 | 30-62.9 | 0.2 | DAC | unspecified | 76-487 | FTIR [a] | Panero et al. 2003 |
| 1227 | 25 | saturated | unspecified | unspecified | 3000* | DFT | Panero and Stixrude, 2004 |
| 1500 | 15 | saturated | LVP | 0-2.95 | 3-456 | FTIR [a] | Bromiley et al., 2006 |
| 1800 | 20-25 | saturated | LVP | 0-7.62 | 25-3010 | FTIR [b] | Litasov et al., 2007 |
| 800-1240 | 8-12.3 | 1-4 | LVP | trace | 53-187 | FTIR [c] | Thomas et al., 2009 |
| 450-550 | 10 | saturated | LVP | 0 | 0.9-1.75 wt. % | TGA/SIMS | Spektor et al., 2011 |
| 627-1654 | 12 | 0-1.73 | LVP | 0-2.24 | 108-1354 | FTIR [b] | Yoshino et al., 2014 |
| 350-550 | 10 | unspecified | LVP | 0 | 0.5-3 wt. % | TGA | Spektor et al., 2016 |
| 450 | 9 | unspecified | LVP | 0 | 3.2 wt. % | LP | Nisr et al., 2017 |
| 950 | 8.4-9.1 | 1.5-6 | LVP | unspecified | 246-376 | FTIR [c] | Frigo et al., 2019 |
| 967-1562 | 32.5-52 | unspecified | DAC | 0 | 4.6-9.9 wt. % | From Nisr et al., 2017 | Lin et al., 2020 |
| 1700 | 20 | saturated | LVP | 3.43-5.37 | 2500-2700 | FTIR [b] | Zhang et al., 2022 |
| 1700 | 28 | saturated | LVP | 4.36 | 2700 | FTIR [b] | Ishii et al., 2022 |
| 973-1592 | 53-72.5 | 0.5-15.2 | DAC | 0 | 0.62-3.61 wt. % | DFT | Lin et al., 2022 |
| 450 | 9 | 8 | LVP | 0 | 1.57-2.25 wt. % | Mass balance/LOI average | Kueter et al., 2023 |

| 1627 | 23 | saturated | LVP | 0 | 0.43-0.62 wt. % | nano-SIMS/NMR | Li et al., 2023 |

Note: DAC, Diamond anvil cell; LVP, large volume press; TGA, Thermogravimetric analysis; SIMS, Secondary ion mass spectrometry, LP, water content determined from lattice parameters, DFT, density functional theory calculation, nano-SIMS/NMR, nanoscale secondary ion mass spectroscopy and solid-state nuclear magnetic resonance.

The calculation methods of water contents based on FTIR are from [a] Pawley et al. (1993), [b] Paterson (1982) and [c] Libowitzky and Rossman (1997).

As shown in Table 1, water solubility in stishovite is significantly influenced by temperature, pressure, and Al content, with large variations from a few wt. ppm to a few wt. percent. However, even under specific pressure conditions, water contents obtained by different studies can differ by more than one order of magnitude. For example, at 20-25 GPa, some studies obtained 25-2700 wt. ppm water (Litasov et al., 2007; Zhang et al., 2022b), while others reported 0.5-3 wt. % water (Spektor et al., 2011; 2016). These discrepancies may partially be attributed to different water content measurement methods. In Table 1, most FTIR studies obtained water contents at the wt. ppm level, while other methods yielded water contents at the wt. % level. For example, Nisr et al. (2017, 2020) suggested a calibration based on unit cell volume dependency on water in stishovite from DFT calculations. Lin et al. (2020) applied  Nisr et al. (2017)'s formulation to calculate $H_2O$ contents in their high-pressure samples resulting in exceptionally high $H_2O$ concentrations exceeding 10 wt. % in many instances. However, Lin et al. (2022) noted that estimating $H_2O$ contents using unit cell volumes from samples synthesized in either the LVP or DAC is highly uncertain.

Additionally, the discrepancies may relate to variations in experimental conditions of temperature, and Al content in stishovite, as detailed in Sect. 4.

### 3.2 Hydrogen incorporation mechanisms

Previous studies indicate two primary mechanisms for water incorporation in stishovite: (1) "hydrogarnet" substitution by $4H^+ \rightarrow Si^{4+}$; and (2) coupled substitution of $H^+$ and $Al^{3+}$ for $Si^{4+}$ ($Al^{3+} + H^+ \rightarrow Si^{4+}$). Hydrogarnet substitution is a common mechanism in pure stishovite. Studies by Litasov et al. (2007) have shown that stishovite can dissolve up to 5 wt. % $Al_2O_3$, which can further increase to 9 wt. % with the presence of water (Ono, 1999). Numerous high-pressure experiments demonstrate that Al can directly couple with H by substituting Si to increase water solubility in Al-bearing stishovite, with the coupled substitution mechanism of $Al^{3+} + H^+ \rightarrow Si^{4+}$ (Pawley et al., 1993; Chung and Kagi, 2002; Bromiley et al., 2006; Lakshtanov et al., 2007a). In addition to $Al^{3+}$, stishovite contains minor amounts of other trivalent cations such as $B^{3+}$, $Fe^{3+}$, $V^{3+}$, and $Cr^{3+}$, which can also facilitate H incorporation into the stishovite structure by similar mechanisms as $Al^{3+}$ (Irifune and Ringwood, 1993; Pawley et al., 1993). However, Litasov et al. (2007) found that $H^+$ (OH) can only co-replace $Si^{4+}$ with up to 40% $Al^{3+}$, which makes the Al/H ratio in stishovite far greater than 1/1, that is, Al in stishovite is much higher than H. Bromiley et al. (2006) also reported excess Al in stishovite, not charge balanced by hydrogen. Thus, the incorporation of aluminum cannot be explained solely by the charge-coupled substitution with protons in stishovite and other substitution mechanisms

have also been proposed. For instance, most Al will occupy the oxygen vacancy ($O_V$) and balance the charge by $2Al^{3+} + O_V^{2+}$ $\rightarrow 2Si^{4+}$ (Pawley et al., 1993; Chung and Kagi, 2002; Litasov et al., 2007; Zhang et al., 2022b).

     Recently, molecular water ($H_2O$) has been identified in stishovite (Lin et al., 2020, 2022; Kueter et al., 2023). The incorporation mechanism is summarized as interstitial $H_2O$ substitution (Lin et al., 2022). Li et al. (2023) presented a new mechanism (one-dimensional (1D) water channels) for molecular water incorporation into stishovite using high-dimensional

neural network (HDNN) potential. Both mechanisms for incorporation of molecular water could explain the weight percent level water observed in Al-free stishovite. Notably, the initial material in these studies is pure $SiO_2$ which means H is impossible coupled with Al to substitute Si. Therefore, even though multiple substitution mechanisms exist for H incorporation in stishovite, the dominant mechanisms in the natural environment remain highly uncertain (Lin et al., 2022). Due to stishovite in subducting slabs contains Al as mentioned above (e.g., Litasov et al., 2007), we emphasize that $Al^{3+} + H^+ \rightarrow Si^{4+}$ is still

very important (e.g., Pawley et al., 1993; Chung and Kagi, 2002; Lakshtanov et al., 2007a; Zhang et al., 2022b; Ishii et al., 2022).

## 4 Factors Dominating Water Solubility in Stishovite

### 4.1 Temperature and Pressure

Numerous experiments have constrained the water solubility in Al-bearing stishovite as a function of temperature and pressure.

Panero et al. (2003) showed that at 30-38 GPa, water contents in stishovite significantly increase with increasing temperature from ~100 wt. ppm at 1500 K to 400 wt. ppm at 3500 K. Litasov et al. (2007) measured the hydrogen contents of stishovite samples synthesized at 20-25 GPa and 1200-1800 °C from several starting materials containing up to 10 wt. % $Al_2O_3$ and found that $H_2O$ contents are positively correlated with temperature between 1000 and 1200 °C, but decrease with increasing temperature at 1200-1800 °C. This can be simply explained by the fact that hydrogen, as an incompatible element, preferably

dissolves into the melt (Smyth, 2006; Litasov et al., 2007). Therefore, the occurrence of melt at high temperature can significantly reduce $H_2O$ contents in stishovite. Panero et al. (2003) showed that at 10-60 GPa, water solubility in stishovite increases with increasing pressure. Panero and Stixrude (2004) suggested that the water solubility in stishovite is ~0.3 wt. % $H_2O$ at 25 GPa and 1500 K, and increases to 1.15 wt. % $H_2O$ at 60 GPa, indicating increasing solubility with pressure. By comparing previous studies, it is evident that water solubility in Al-bearing stishovite shows an overall weak positive

correlation with temperature but a significant decrease when the melt is present at >1600 °C. Whereas, the pressure dependence is more complex, with a positive correlation below 20 GPa but a negative correlation above 20 GPa, suggesting an optimal water solubility in stishovite at ~20 GPa (Fig. 4). In addition, Lakshtanov et al. (2007b) suggested that stishovite undergoes a transition to $CaCl_2$-structured phase (post-stishovite) at ~30 GPa. Weight percent levels of water (0.85 to 1.1 wt. %) in $CaCl_2$-structured Al-rich post-stishovite at 24 to 28 GPa and 1000 to 2000 °C have been reported by Ishii et al. (2022). This implies

stishovite can retain its water content through phase transition with increasing pressure. However, Fischer et al. (2018)

suggested the transition to post-stishovite occurs at pressures of 68–78 GPa in the Earth. Therefore, if the post-stishovite can accommodate water at ~20 GPa where the Al-bearing stishovite has a peak water solubility remains unclear.

## 4.2 Al Content

Given that natural systems always contain some Al, its effect on water solubility in stishovite has been a focus of research. Pawley et al. (1993) found that for stishovite with >1 wt. % $Al_2O_3$, the $H_2O$ content (82 wt. ppm) is ten times higher than Al-free stishovite (7 wt. ppm). Chung and Kagi (2002) showed that at 15 GPa and 1400 °C, with increasing trivalent cations (mostly Al) from 0.612 wt. % to 1.341 wt. %, $H_2O$ contents in stishovite increase from within the range of 128 wt. ppm to 536 wt. ppm. Litasov et al. (2007) obtained the maximum 3010 wt. ppm $H_2O$ in Al-bearing stishovite (4.4 wt. % $Al_2O_3$) versus 16-30 wt. ppm $H_2O$ in Al-free stishovite at 20 GPa and 1400 °C. Recent work by Ishii et al. (2022) also demonstrated the positive correlation between water solubility and $Al_2O_3$ content in stishovite. Therefore, increasing Al significantly enhances water solubility in stishovite. Previous data demonstrates a positive correlation between Al and water contents in stishovite (Fig. 5).

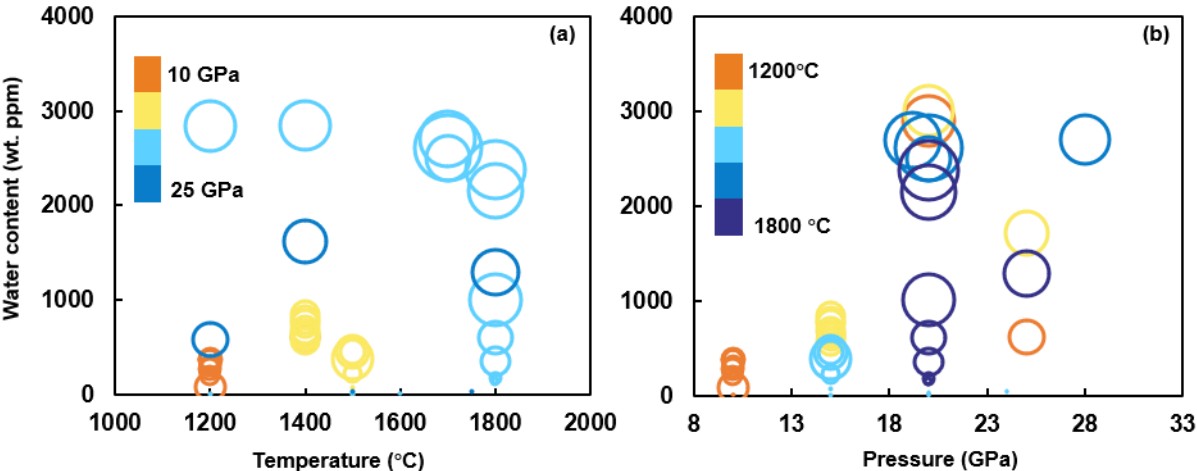

**Figure 4: Previous studies on water solubility in Al-bearing stishovite. The dimension of circles represents the Al content. (a) Water content versus temperature; (b) Water content versus pressure (Only FTIR data are presented (Data from Pawley et al., 1993; Chung and Kagi, 2002; Bromiley et al., 2006; Litasov et al., 2007; Ishii et al., 2022; Zhang et al., 2022b). The relation of water solubility and Al content is shown in Fig. 5.**

In contrast, Lin et al. (2022) showed decreasing water solubility in pure stishovite with increasing temperature and pressure, markedly different from Al-bearing systems. Also, Lin et al. (2022)'s experimental data indicates a $H_2O$ storage capacity in Al-free stishovite of ~3.5 wt. % at ~50 GPa and 1800 K. Kueter et al. (2023) reported the Al-free stishovite contains on average 1.69 wt. % water at 9 GPa and 450 °C. However, previous LVP studies synthesized Al-free (or Al-poor at the wt. ppm level) stishovite under water-saturated conditions is nearly anhydrous and commonly contains less than 100 wt. ppm $H_2O$, which is significantly different from wt. % $H_2O$ in recent studies (e.g., Spektor et al., 2011, 2016; Lin et al., 2022; Kueter et

al., 2023). It is challenging to understand how different studies can produce such widely disparate results, with differences of several orders of magnitude for the $H_2O$ capacity of stishovite (Lin et al., 2022). The reason for the difference is unclear and there are various possible explanations. One possible cause is water loss from the capsule (e.g., Litasov et al., 2007). Moreover, the large discrepancy of water solubility between Al-stishovite (wt. ppm level) and Al-free stishovite (wt. % level from recent studies) has partly been explained by a hydrogarnet substitution mechanism ($Si^{4+} \leftrightarrow 4H^+$) and/or the incorporation of interstitial molecular water (Kueter et al., 2023). However, the discrepancy caused by different measurement methods should be taken into consideration as well.

It should be noted that the behavior of Al itself also depends on pressure and temperature. Al can remarkably increase the hydrogen solubility of stishovite (Fig. 5), hydrogen can also increase Al solubility of stishovite in return. However, previous studies mainly focus on the water solubility of Al-free or Al-saturated stishovite. Studies on the P-T dependence of Al solubility in stishovite are very limited (e.g., Liu et al., 2006; Litasov et al., 2007). Liu et al. (2006) investigated Al solubility in dry stishovite in anhydrous experiments at 15-25 GPa and 1350 °C-2150 °C. Liu et al. (2006) found that $Al_2O_3$ solubility in dry stishovite is slightly but consistently reduced by pressure increase, however, its response to temperature increase, is more complicated: increases at low temperatures, maximizes at around 2000 °C, and perhaps decreases at higher temperatures. Therefore, quantitatively constraining water solubility in natural stishovite requires considering Al solubility under different P-T conditions. In summary, the role of Al and other impurities in the water content and stability of hydrous stishovite remains poorly understood (Kueter et al., 2023).

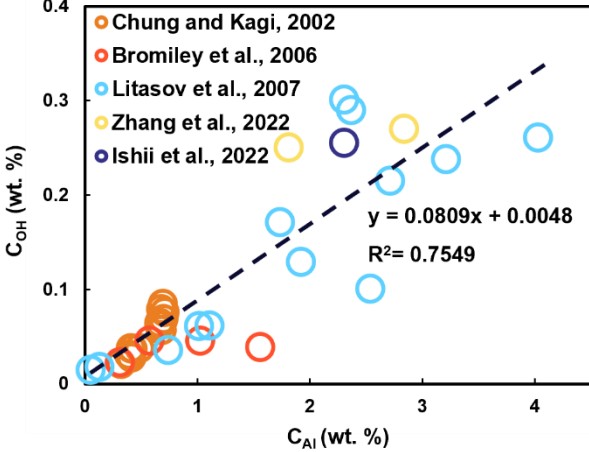

**Figure 5: The correlation between $H_2O$ and $Al_2O_3$ in stishovite. Data are from Chung and Kagi (2002), Bromiley et al. (2006), Litasov et al. (2007), Ishii et al. (2022), and Zhang et al. (2022b), respectively. In Al-free stishovite, water contents range from 0.0003-0.0108 wt. % (3-108 wt. ppm), In Al-bearing stishovite, water contents increase to 0.0082-0.301 wt. % (82-3010 wt. ppm) with increasing Al.**

## 4.3 Other Factors

Apart from aluminum (Al), other impurity ions likely play a role in affecting water solubility in stishovite. Several studies have suggested (Ono et al., 2002; Chung and Kagi, 2002; Panero et al., 2003; Panero and Stixrude, 2004) that stishovite formed from subducted MORB might incorporate a variety of trivalent cations, such as $Al^{3+}$, $Cr^{3+}$, $Fe^{3+}$, $V^{3+}$ and $Ti^{3+}$. These cations could potentially increase water content through coupled substitution. However, the work of Litasov et al. (2007) reported a small discrepancy in the $H_2O$ content between Fe-bearing stishovite (likely containing $Fe^{3+}$) and Al-free and Fe-free stishovite. Consequently, the implications of $Fe^{3+}$, $Ti^{3+}$, and other elements on water solubility in stishovite need further investigation.

In addition, oxygen fugacity ($fO_2$) stands as a crucial thermodynamic parameter that describes the oxidation state in deep Earth. It has a strong impact on the water solubility of those NAMs containing valency elements (e.g., Fe and Ti). For instance, the water solubility in olivine and garnet displays positive and negative correlations with $fO_2$, respectively (Yang, 2016; Zhang et al., 2022a). Since a small amount of Fe and Ti determined in a pseudomorph of former stishovite from a continental subduction slab (Liu et al., 2018), $fO_2$ could theoretically affect the water solubility of stishovite. However, the $fO_2$ dependence on water solubility of stishovite remains unknown and requires further investigation to provide clear insights.

## 5 Water solubility in stishovite as a function of pressure, temperature, water fugacity, and Al content

Numerous studies have revealed that the water content in NAMs basically follows a thermodynamic relationship with temperature, pressure and water fugacity as shown in Eq. (1) (e.g., Kohlstedt et al., 1996; Zhao et al., 2004; Karato, 2010):

$$C_{OH} = A \cdot f_{H_2O}^n \cdot exp\left(-\frac{\Delta E + \Delta V \cdot P}{R \cdot T}\right), \qquad (1)$$

where $C_{OH}$ is the water content, $A$ is a constant, $f_{H_2O}^n$ is the water fugacity, the value of n depends on the hydrogen incorporation mechanism, and $\Delta E$ and $\Delta V$ denote the energy and volume changes associated with hydrogen dissolution.

Lin et al. (2022) suggested a relationship for water solubility in Al-free stishovite as a function of pressure and temperature. Considering the stishovite in a subducted slab contains Al up to 5% (Litasov et al., 2007), and the remarkable influence of aluminum (Al) on hydrogen solubility of stishovite ($Al^{3+} + H^+ \rightarrow Si^{4+}$), therefore, we here propose a new model for water solubility in Al-bearing stishovite as Eq. (2):

$$C_{OH}^{St} = A \cdot f_{H_2O}^n \cdot exp\left(-\frac{\Delta E + \Delta V \cdot P}{R \cdot T}\right) \cdot exp\left(\frac{B \cdot X_{Al}}{R \cdot T}\right) \qquad (2)$$

In Eq. (2), $C_{OH}^{St}$ is water solubility in Al-bearing stishovite, $B$ is a constant, and $X_{Al}$ is the molar fraction of Al in stishovite. Given that the dominant hydrogen incorporation mechanism corresponds to $Al^{3+} + H^+ \rightarrow Si^{4+}$ (Pawley et al., 1993), yielding $C_{OH}^{St} \propto f_{H_2O}^{0.5}$, thus $n = 0.5$ (Kohlstedt et al., 1996; Karato, 2010).

To eliminate large disparities arising from diverse analytical methods as discussed above, we excluded the published water solubility in Al-bearing stishovite obtained by methods such as TGA/SIMS and only collected $H_2O$ solubility from FTIR

measurements (Chung and Kagi, 2002; Litasov et al., 2007; Ishii et al., 2022; Zhang et al., 2022b, as shown in Table 1). We used a non-linear least square fitting method to fit Eq. (2) and obtained parameters as follows: $n = 0.5$, $A = 24\pm13$ ppm/GPa$^{0.5}$, $\Delta E = -3.06\pm0.88$ kJ/mol, $\Delta V = 4.29\pm0.27$ cm$^3$/mol, $B = 7.69\pm1.12$ kJ/mol. The uncertainty is one standard deviation. As shown in Fig 6, a very good correlation can be found between the experimental data and the calculated results from Eq. (2) (Fig. A1).

Due to the data we used to fit Eq. (2) are from experiments carried out at wide conditions of 10-28 GPa and 1200-1800 °C (close to the condition of MTZ), therefore, Eq. (2) can now reasonably estimate the water solubility in Al-bearing stishovite across temperature, pressure, and $f_{H_2O}$ conditions of MTZ.

We conducted calculations to determine the water solubility in stishovite containing 3 mol. % aluminum at temperatures ranging from 800 °C to 1600 °C and pressures from 10 GPa to 40 GPa (Fig. 6). The results indicate that up to 30 GPa, water content experiences a marginal decline as temperature rises, while beyond 30 GPa, a positive correlation emerges between water content and temperature. This could be attributed to a lowered solidus under reduced pressures, leading to diminished water solubility as high-temperature, water-rich melts form. Conversely, at pressures surpassing 30 GPa, the heightened solidus counteracts the melt effect. Moreover, the solubility of water experiences a marked increase at pressures below 22 GPa to 32 GPa, followed by a decrease at pressures beyond this range, signifying an optimal solubility window. This aligns well with the overarching experimental data trend (Fig. 6b).

## 6 Implications for water transport to the deep Earth

Subduction zones serve as the exclusive pathway for surface water to infiltrate the Earth's deep interior. The transport and quantities of water within subducting slabs have profound impacts on surface environments, deep Earth properties and dynamics (Shillington, 2018). Existing experimental and geophysical data suggest the mantle transition zone distributes potentially water-rich regions with >1 wt. % H$_2$O (Pearson et al., 2014). However, the origins of this water remain a subject of debate. One perspective posits that the deep mantle has retained intrinsic wetness since Earth's formation. Alternatively, NAMs in subducting slabs may carry a significant amount of water to the deep mantle (Peslier et al., 2017). However, it is essential to note that most hydrous minerals within subducting slabs will break down at various depths, typically having stability limits below 9 GPa, which limits the transport of water to greater mantle depths (>300 km) (Zheng et al., 2016). For instance, minerals like antigorite/phlogopite and lawsonite, which can survive to maximum depths in cold slabs, tend to dehydrate at around 300 km (Poli and Schmidt, 2002; Walter, 2021). Therefore, a fundamental question arises: can the released water from these hydrous minerals be further carried to even greater depths by NAMs?

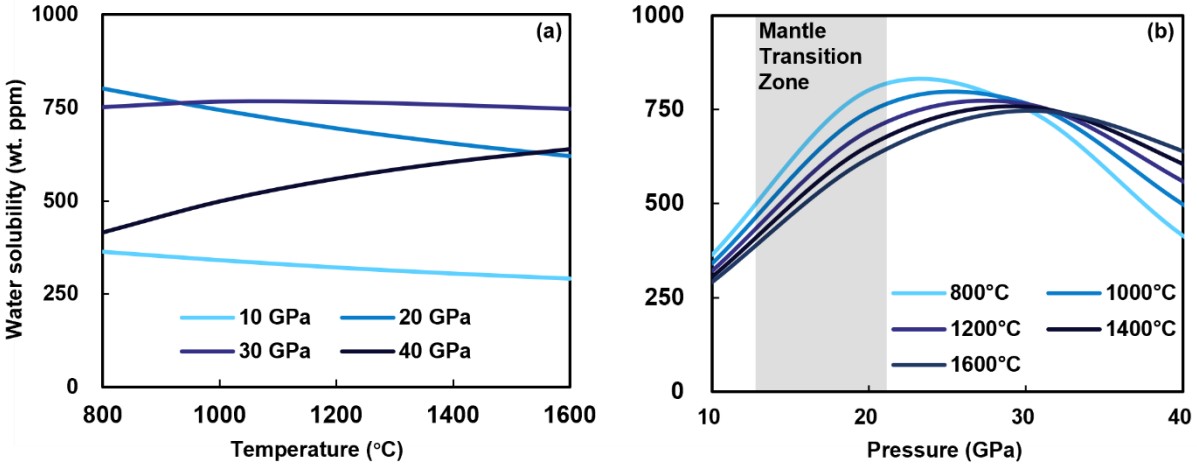

**Figure 6: Water solubility in stishovite versus (a) temperature and (b) pressure**

Existing research demonstrates that water storage capacities within upper mantle NAMs generally increase with depth (Yang and Li, 2016). Nevertheless, it is imperative to investigate how phase transitions and breakdown reactions of NAMs influence the transport of water in subducting slabs at greater depths. For instance, Gavrilenko (2008) showed that water solubility in
clinopyroxene (Cpx) decreases along a mantle geotherm (Litasov and Ohtani, 2007) from 517 wt. ppm at 10 GPa to 176 wt. ppm at 16 GPa. Given the concurrent decomposition of Cpx from approximately 65 vol. % at 9 GPa to 0 vol. % at 16 GPa, an estimated 310 wt. ppm of water would be released from Cpx within the subducted oceanic slab.

We further conducted calculations to assess the evolution of water storage capacities in major minerals within subducted MORB and continental sediments along a cold subduction slab geotherm suggested by Litasov and Ohtani (2007). The results
indicate that garnet and stishovite serve as primary carriers of water, with higher water storage capacities observed in subducted oceanic crust compared to sediments (Fig. 7). In subducted MORB and continental sediments down to mantle depths, stishovite comprises 10-20 vol. % (Irifune et al., 1986; Ono et al., 2001) and >20 vol. % (Ishii et al., 2019), respectively. Our model demonstrates that stishovite water solubility increases from 998 wt. ppm at 14 GPa to 1317 wt. ppm at 22 GPa along the cold subduction slab geotherm (Fig. 7). Consequently, stishovite may theoretically accommodate water released from the
breakdown of hydrous minerals (Lin et al., 2020; Nisr et al., 2020; Walter, 2021). Due to a slight negative temperature dependence on water solubility of stishovite at 20 GPa (Fig. 6a), the water in stishovite will decrease with increasing temperature. That indicates a water outflow from stishovite along with the subducted slab being heated in MTZ. Comparison with a hot subduction geotherm (Table S2 and Fig. B1), slabs will reduce more water in a cold subduction geotherm. Additionally, stishovite water solubility peaks at 22-32 GPa (corresponding conditions of MTZ or topmost of the lower mantle)
(Fig. 6) and declines at higher pressure conditions. Previous studies suggest hydration of a large region of the transition zone and that dehydration melting may act to trap $H_2O$ in the transition zone (e.g., Schmandt et al., 2014). Taking into account the

occurrence of melts, the optimal solubility likely falls within the water-rich mantle transition zone at pressures below 22 GPa. Consequently, stishovite emerges as a key transporter supplying water to the mantle transition zone (Walter, 2021).

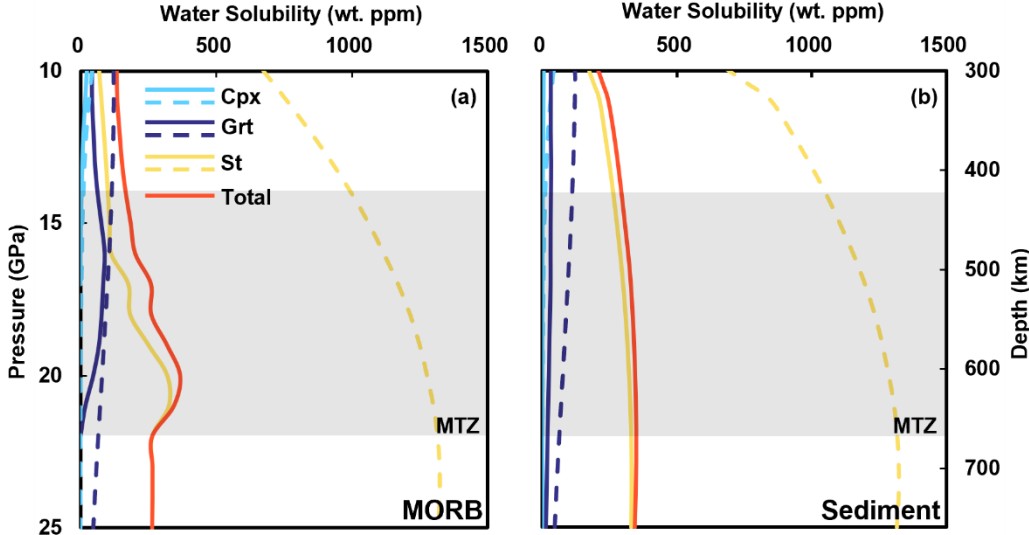

**Figure 7: Water solubility in a cold subducted slab. Dashed lines represent water solubility of minerals and solid lines show the actual water contents considering mineral modal fractions (from Litasov and Ohtani, 2007) in subduction slabs. A cold subduction geotherm gradient from Litasov and Ohtani (2007) is used for the calculation (Table S2). The calculated result along with a hot subduction geotherm is presented in Fig. B1 in the Appendix. Water solubility models: Grt from Lu and Keppler (1997), Cpx from Gavrilenko (2008), St from this study.**

## 7 Conclusions and outlook

To sum up, the comprehensive review and exploration in this paper have encompassed water solubility in stishovite, hydrogen incorporation mechanisms and governing factors. We've constructed a new model for water solubility in Al-bearing stishovite and delved into stishovite's role in transporting water through a subducting slab and implications for water distribution in deep Earth. The key conclusions can be summarized as follows:

1. Water solubility in stishovite exhibits a positive correlation with Al content, which increases water concentrations via coupled substitution $Al^{3+} + H^+ \rightarrow Si^{4+}$, as well as enhances the incorporation of interstitial $H_2O$ by Al substitution-induced expansion of the stishovite structure.

2. The relationship between water solubility of Al-bearing stishovite and water fugacity, temperature, pressure, and Al content can be described as $C_{OH}^{St} = A \cdot f_{H_2O}^n \cdot exp\left(-\frac{\Delta E + \Delta V \cdot P}{R \cdot T}\right) \cdot exp\left(\frac{B \cdot X_{Al}}{R \cdot T}\right)$ with $n = 0.5$, $A = 24 \pm 13$ ppm/GPa$^{0.5}$, $\Delta E = -3.06 \pm 0.88$ kJ/mol, $\Delta V = 4.29 \pm 0.27$ cm$^3$/mol, $B = 7.69 \pm 1.12$ kJ/mol.

        3. The calculation results from Eq. (2) demonstrate that stishovite's water solubility strongly depends on pressure, exhibiting a positive correlation below 22-32 GPa and a negative correlation above, indicating a peak solubility range at 22-32 GPa.
While temperature dependence is weakly negative up to 30 GPa and weakly positive beyond.

        4. Following a cold subduction slab geotherm, stishovite's water solubility increases from 998 wt. ppm at 14 GPa to 1317 wt. ppm at 22 GPa. This suggests that stishovite can potentially accommodate water released from the breakdown of hydrous mineral, contributing to the water-rich transition zone.

        However, several crucial unanswered questions persist in research on water solubility of stishovite : 1. The solubility of Al
and its impact on water solubility across varying pressure-temperature conditions requires clarification; 2. The roles of minor elements like Fe and Ti, as well as oxygen fugacity, remain unclear; 3. Partitioning coefficients governing water distribution between stishovite and other major phases (e.g., Cpx, Grt, melts)  during slab subduction are not yet understood; 4. Equation (2) proposed in this paper needs refinement through systematic experiments conducted under controlled conditions. These issues are essential for understanding how water transports into deep Earth during subduction and require further detailed
experimental and simulation-based investigations to address.

## Appendix A: Comparison between experimental and calculated results

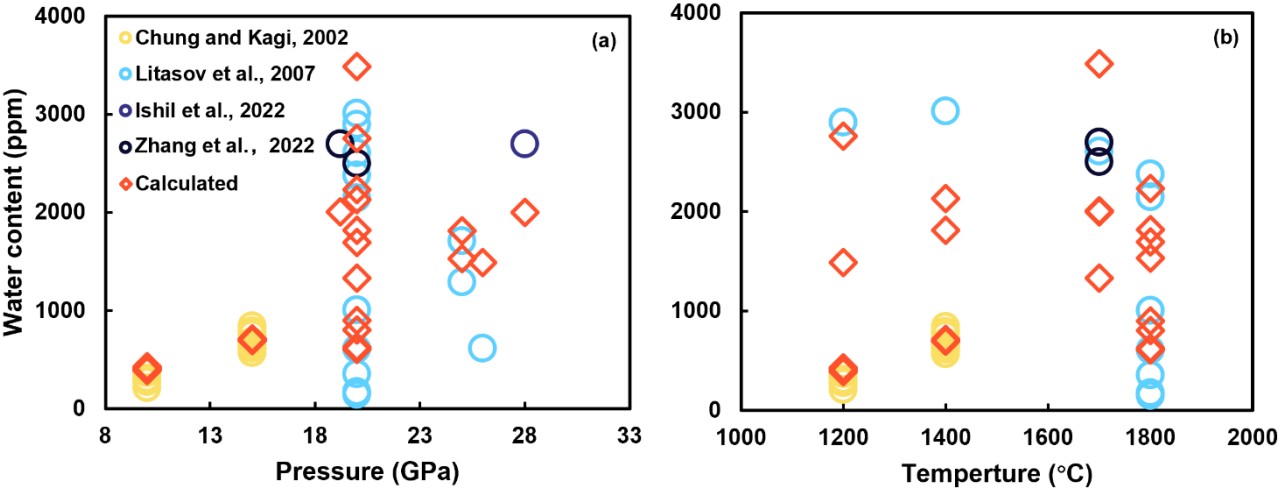

**Figure A1: Comparison between experimental and calculated results (Table S1).**

**Appendix B: Water solubility of minerals in hot subducted slabs at 10-25 GPa**

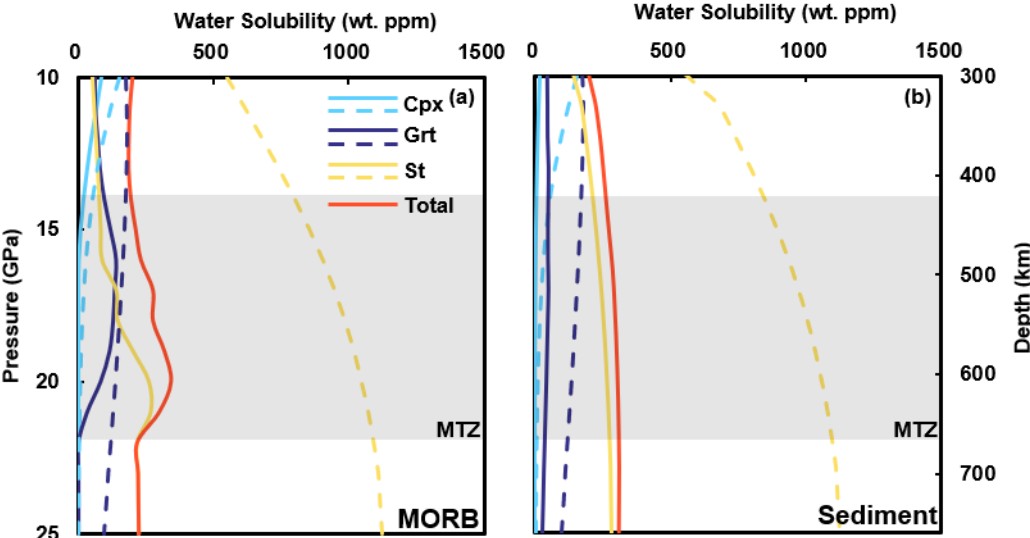

**Figure B1: Water solubility in hot subducted slabs. Dashed lines represent water solubility of minerals and solid lines show the actual water contents considering mineral modal fractions (from Litasov and Ohtani, 2007). Water solubility models: Grt from Lu and Keppler (1997), Cpx from Gavrilenko (2008), St from this study.**

*Data availability.*  Data have been made available in the Supplement.

*Author contribution.*    Mengdan Chen: Data curation, formal analysis and writing-original draft. Lei Kang: Writing - review & editing, foundation support. Danling Chen and Liang Liu: Writing - review & editing. Changxin Yin and Long Tian: Review and language modification.

*Competing interest.*  The authors declare that they have no conflict of interest.

*Acknowledgments.*  We are grateful to Professor Shun Karato and Dr. Xin Li for the discussion on Al behavior in stishovite. Thanks to three anonymous reviewers for their constructive comments that significantly improved this article.

*Financial support.*  This study is supported by the Natural Science Foundation of China (Grant No. 42030307, 41972058, 42372064) and the MOST Special Fund from the State Key Laboratory of Continental Dynamics, Northwest University (Grant No. 201210233).

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
