# Peer review of "Hydrogen solubility of stishovite provides insights into water transportation to deep Earth"

_EGUsphere, 2023_

## Referee Comment (RC1)

Referee comment on Chen et al, "Hydrogen solubility of stishovite provides insights into water transportation to deep Earth."

**General Comment**

The Authors present an empirical equation about the water solubility in Al-bearing stishovite that is based on literature data provided by experimental studies. The equation is used to model the water concentrations of Stv and bulk MORB crust along a subduction geotherm, showing that the water capacity increases till the base of the transition zone and then sharply declines.

There are a few important issues (detailed below) that need to be addressed. The most important ones are an insufficient discussion on the derivation of the equation and its limits, and the missing discussion on the presence of molecular water in stishovite.

The manuscript is well-structured and mostly clearly understandable, although it appears a bit superficial. This contradicts the authors' claim to present a "comprehensive review". While indeed many important papers are cited, I feel that the work *with* the papers is rather shallow. A comprehensive review needs a more in-depth analysis and more critical data assessment.

The new aspect of the manuscript is the abovementioned equation. Similar models exist (e.g. work of Lin and coworkers cited in the manuscript), and I am unsure if the presented work provides significant new insights.

I think a more thorough revision is required.

Kind regards

**Technical and specific comments:**

**Chapter 1**

Line 22 – "Explicitly and implicitly". While the meaning is explained, these terms are not usually used in the field (and not further in the manuscript) and remain somewhat ambiguous. I would suggest finding a different phrasing or description, e.g., molecular and structurally bound water.

Line 34 -  "Although…." sentence needs references.
Line 61 – "preservation conditions of Stishovite are very harsch …", the word "harsh" is a bit odd and not very descriptive. I suggest rephrasing it to something like "Stv is unstable and easily transforms to lower pressure $SiO_2$ polymorphs."

Figure 1 -  While cited correctly, this figure is the original figure of Litasov & Ohtani (2007), simply cropped to smaller dimensions. It's not "after", which would imply redrawing or significant modifications. I don't think that is okay, as it implies own artwork.

**Chapter 3 – Water solubility and incorporation mechanisms in stishovite.**

Generally, for a review, I found this section very superficial. A more in-depth literature study would be beneficial.

Section 3.1

Line 90 – "As shown in Table 1, …". Table 1 lists literature values on synthesis conditions and water contents. Other than stated, it is not directly apparent from the table that correlations exist between water concentration and experimental and chemical parameters. This needs to be elaborated in more detail and displayed in figures (as it has been shown for Al and $H_2O$ concentrations in Fig 5). I don't find Fig 4 very helpful, as it compresses (at least) 4 variables into one plot.

Line 91 – "…differ by more than one order of magnitude". Kueter et al 2023 (CMP) discuss discrepancies in the water content between Al-free and Al-bearing Stv, as well as for Al-free Stv synthesized in DAC and Multi Anvil setups. This review should note & discuss this as well.

Table 1 – The table would benefit from additional information about the synthesis method (e.g., Multi Anvil, Diamond anvil cell). Please also specify "other methods" annotated with an asterisk. Please add the most recent work as well.

Section 3.2

This section is about the hydrogen incorporation mechanism in Stv. Two mechanisms are mentioned, i.e., hydrogarnet and Al-H substitutions. Recent studies also found evidence for molecular water ($H_2O$) in the stishovite (Lin et al, 2020 & 2022; Kueter et al 2023; Li et al 2023). I think the Authors briefly mention $H_2O$ as "explicit water" in the introduction (if I understand this correctly), but it's not further discussed. The apparent presence of molecular water is important and should be discussed as well. Particularly the Li et al 2023 study is interesting, as it suggests that water is not homogeneously distributed but aligns in one-dimensional $SiO_2$-H2O superclusters.

**Chapter 4**

L 117 – "Numerous..". I don't think the sentence is necessary. It contains no information.

L123 – "melt proportions". Please give a few details about Litasov experiments. What kind of melt?

Section 4.1

The effect of water solubility of Stv to CaCl2-structured Stv should be discussed here as well (e.g. Ishii et al 2022). The DAC experiments and models of Lin (2020, 2022) should also be considered.

Section 4.2.

As mentioned above, Al content seems to have a positive correlation with $H_2O$ content in the 100 to 1000[th] ppm level. However, Al-free Stv incorporates wt% amounts of $H_2O$; something that's not readily seen with Al-bearing Stv. The manuscript would benefit from discussing this a bit more. See, for example, Kueter et al 2023.

Line 144 – "Statistically compiling…" I don't understand the wording. Figure 5 does not show any statistics or statistical breakdown of the data. It's a summary plot of available data. Also, please cite the literature shown in Fig 5. I would recommend being more quantitative as well: Maybe you can provide a regression of the data and plot it in Fig 5 (similar to what Litasov et al. 2007 did).

Line 167 onward - The fO2 part of section 4.2 is too vague and qualitative. There is no real information gain from it. I would recommend to discussing it in more detail, or (less recommended) omitting it.

**Chapter 5 – Water solubility…**

The following is, in my opinion, the most important issue that has to be addressed:
This chapter is the core of the study and subsequent models rely on equation 2 given in it. Consequently, Eqn 2 needs to be carefully derived and explained:

It is important to
  - discuss the dismissal and use of literature experimental data in more detail
  - show a plot that displays the data and the fits from which Eqn 2 has been derived
  - make a thorough error/uncertainty analysis of the fit
  - discuss the obtained fitting parameters (are they realistic, do they compare with literature?)
  - discuss limits of the fitting method (e.g. boundaries, uncertainties)

I further would recommend providing at least one example calculation (main text or appendix). This helps the reader tremendously to comprehend and reproduce the data & models. Also, please provide the exact literature data in the appendix that you used for the fitting.

Since the study focuses on Al-Stv only, it is ok to omit the high $H_2O$ experiments of Al-free Stv, but it would be good to mention this in the text and give reasons for it.

Last but not least: Eqn 2 has a typo ("epx", same in the conclusion part)

**Chapter 6**

The second half of the first paragraph should also cite Walter (2021, Natl Sci Rev, 2021, Vol. 8).

Fig. 6 - I am not really sure what Fig. 6a is based on. Also, range is 800-1400 °C, caption says 1000-1400 °C). 6b – the curve maxima are basically the Stv to CaCl2-Stv phase transitions? Please explain in more detail. Maybe mark the transition interval in the diagram.

L 224 – Please refer to fig. 7.

L 225 – "Consequently, Stishovite…" The statement in this sentence is correct, but not very new. Several previous studies concluded this, e.g. Walter 2021 and refs within. That should be cited accordingly.

L 228 – "Taking into …" Please elaborate on the melts and outline your reasoning a bit more in detail.

Figure 7 – "Yin and Kang, 2023 (in preparation). I disagree with citing non-peer reviewed work. Is there an alternative?

**Paper mentioned:**

Walter, M. J. (2021). Water transport to the core–mantle boundary. *National Science Review*, *8*(4), nwab007.

Nisr, C., Chen, H., Leinenweber, K., Chizmeshya, A., Prakapenka, V. B., Prescher, C., ... & Shim, S. H. (2020). Large H2O solubility in dense silica and its implications for the interiors of water-rich planets. *Proceedings of the National Academy of Sciences*, *117*(18), 9747-9754.

Li, J., Lin, Y., Meier, T., Liu, Z., Yang, W., Mao, H. K., ... & Hu, Q. (2023). Silica-water superstructure and one-dimensional superionic conduit in Earth's mantle. *Science Advances*, *9*(35), eadh3784.

Litasov, K. D., Kagi, H., Shatskiy, A., Ohtani, E., Lakshtanov, D. L., Bass, J. D., & Ito, E. (2007). High hydrogen solubility in Al-rich stishovite and water transport in the lower mantle. *Earth and Planetary Science Letters*, *262*(3-4), 620-634.

Lin, Y., Hu, Q., Meng, Y., Walter, M., & Mao, H. K. (2020). Evidence for the stability of ultrahydrous stishovite in Earth's lower mantle. *Proceedings of the National Academy of Sciences*, *117*(1), 184-189.

Lin, Y., Hu, Q., Walter, M. J., Yang, J., Meng, Y., Feng, X., ... & Mao, H. K. (2022). Hydrous SiO2 in subducted oceanic crust and H2O transport to the core-mantle boundary. *Earth and Planetary Science Letters*, *594*, 117708.

Kueter, N., Brugman, K., Miozzi, F., Cody, G. D., Yang, J., Strobel, T. A., & Walter, M. J. (2023). Water speciation and hydrogen isotopes in hydrous stishovite: implications for the deep Earth water cycle. *Contributions to Mineralogy and Petrology*, *178*(8), 48.

Ishii, T., Criniti, G., Ohtani, E., Purevjav, N., Fei, H., Katsura, T., & Mao, H. K. (2022). Superhydrous aluminous silica phases as major water hosts in high-temperature lower mantle. *Proceedings of the National Academy of Sciences*, *119*(44), e2211243119.

---

## Referee Comment (RC3)

[referee-annotated manuscript omitted]

---

## Author Comment (AC1)

Referee comment on Chen et al, "Hydrogen solubility of stishovite provides insights into water transportation to deep Earth."

**General Comment**

The Authors present an empirical equation about the water solubility in Al-bearing stishovite that is based on literature data provided by experimental studies. The equation is used to model the water concentrations of Stv and bulk MORB crust along a subduction geotherm, showing that the water capacity increases till the base of the transition zone and then sharply declines.

There are a few important issues (detailed below) that need to be addressed. The most important ones are an insufficient discussion on the derivation of the equation and its limits, and the missing discussion on the presence of molecular water in stishovite.

The manuscript is well-structured and mostly clearly understandable, although it appears a bit superficial. This contradicts the authors' claim to present a "comprehensive review". While indeed many important papers are cited, I feel that the work with the papers is rather shallow. A comprehensive review needs a more in-depth analysis and more critical data assessment.

The new aspect of the manuscript is the abovementioned equation. Similar models exist (e.g. work of Lin and coworkers cited in the manuscript), and I am unsure if the presented work provides significant new insights.

I think a more thorough revision is required.

Kind regards

**Technical and specific comments:**

**Chapter 1**

Line 22 – "Explicitly and implicitly". While the meaning is explained, these terms are not usually used in the field (and not further in the manuscript) and remain somewhat ambiguous. I would suggest finding a different phrasing or description, e.g., molecular and structurally bound water.

Thank you for this suggestion. We replaced "Explicitly and implicitly" with molecular water and structurally bound water in the revised text following your suggestion.

Line 34 - "Although…." sentence needs references.

Thanks. We added the citation (Kaminsky, 2012).

Line 61 – "preservation conditions of Stishovite are very harsch …", the word "harsh" is a bit odd and not very descriptive. I suggest rephrasing it to something like "Stv is unstable and easily transforms to lower pressure $SiO_2$ polymorphs."

Thanks very much for your suggestion. We rephrased this statement as "stishovite is unstable and easily transforms to lower pressure $SiO_2$ polymorphs".

Figure 1 - While cited correctly, this figure is the original figure of Litasov & Ohtani (2007), simply cropped to smaller dimensions. It's not "after", which would imply redrawing or significant

modifications. I don't think that is okay, as it implies own artwork.

Thank you for pointing out this error in the manuscript. We agreed and deleted "after".

**Chapter 3 – Water solubility and incorporation mechanisms in stishovite.**

Generally, for a review, I found this section very superficial. A more in-depth literature study would be beneficial.

Section 3.1

Line 90 – "As shown in Table 1, …". Table 1 lists literature values on synthesis conditions and water contents. Other than stated, it is not directly apparent from the table that correlations exist between water concentration and experimental and chemical parameters. This needs to be elaborated in more detail and displayed in figures (as it has been shown for Al and $H_2O$ concentrations in Fig 5). I don't find Fig 4 very helpful, as it compresses (at least) 4 variables into one plot.

Thank you for pointing out this problem in manuscript. Yes, the values listed in Table 1 are not so clear to figure out the relationship between water content and experimental conditions and chemical compositions. One reason is that different authors carried out their experiments using different apparatus (Multi Anvil, Diamond Anvil Cell) at different conditions (e.g., P, T, Al content). Another reason is different measurement methods of water content in Stv make the relationship puzzled. Therefore, we changed Fig. 4 to a bubble chart in which the dimension of circles represents the Al content.

Line 91 – "…differ by more than one order of magnitude". Kueter et al 2023 (CMP) discuss discrepancies in the water content between Al-free and Al-bearing Stv, as well as for Al-free Stv synthesized in DAC and Multi Anvil setups. This review should note & discuss this as well.

Thank you for the above suggestions. Kueter et al. (2023)'s work is very helpful. We added Kueter et al. (2023)'s result in Table 1 and discussed this work in some sections of our paper.

Table 1 – The table would benefit from additional information about the synthesis method (e.g., Multi Anvil, Diamond anvil cell). Please also specify "other methods" annotated with an asterisk. Please add the most recent work as well.

Thank you very much for this suggestion. We added two columns to Tabel 1, one is "Synthesis method", another is "Measurement method". The most recent works by Kueter et al. (2023) and Li et al. (2023) are also added.

Section 3.2

This section is about the hydrogen incorporation mechanism in Stv. Two mechanisms are mentioned, i.e., hydrogarnet and Al-H substitutions. Recent studies also found evidence for molecular water ($H_2O$) in the stishovite (Lin et al, 2020 & 2022; Kueter et al 2023; Li et al 2023). I think the Authors briefly mention $H_2O$ as "explicit water" in the introduction (if I understand this correctly), but it's not further discussed. The apparent presence of molecular water is important and should be discussed as well. Particularly the Li et al 2023 study is interesting, as it suggests that water is not homogeneously distributed but aligns in one-dimensional $SiO_2$-H2O superclusters.

Thanks very much for this comment. We thought the water of no matter what species ($OH^-$, $H_2O$) in Stv is so-called "implicit water" (we removed this term in the revised text as you suggested) because no

stoichiometric hydron is present in the chemical formula of Stv.

Yes, molecular water ($H_2O$) has been identified in recent works (e.g., Lin et al., 2022). The incorporation mechanism is summarized as interstitial $H_2O$ substitution. However, Li et al. (2023) presented a new mechanism (one-dimensional (1D) water channels) for water incorporation into stishovite using high-dimensional neural network (HDNN) potential. This is another alternative and important mechanism for the incorporation of molecular water in stishovite. We added related discussion in the revised main text.

**Chapter 4**

L 117 – "Numerous…". I don't think the sentence is necessary. It contains no information.

Thanks. We removed this sentence.

L123 – "melt proportions". Please give a few details about Litasov experiments. What kind of melt?

We feel sorry for the problem brought to you. This statement in the main text explains that hydrogen as an incompatible element favors the melt as a result of increasing temperature. Litasov et al. (2007) found the coexisting melt will reduce the water content in stishovite from their experiments. However, Litasov et al. (2007) have not clarified the properties of melt coexisting with stishovite. Nevertheless, we add some details about the experiments by Litasov et al. (2007) in revised text.

Section 4.1

The effect of water solubility of Stv to CaCl2-structured Stv should be discussed here as well (e.g. Ishii et al 2022). The DAC experiments and models of Lin (2020, 2022) should also be considered.

Thank you very much for this suggestion. Ishii et al. (2022) reported high water solubility in $CaCl_2$-structured Stv at 24 to 28 GPa and 1000 to 2000 °C. However, the transition pressure from Stv to $CaCl_2$-structured Stv is debated as Fischer et al. (2018) suggested it can be high as 68-78 GPa in the Earth. Therefore, we add some more discussion on the water in $CaCl_2$-structured Stv but emphasize the pressure and temperature dependence on water solubility of Al-bearing Stv in our paper. The DAC experiments of Lin et al. (2020,2022) are discussed in Section 4.2.

Section 4.2.

As mentioned above, Al content seems to have a positive correlation with $H_2O$ content in the 100 to 1000[th] ppm level. However, Al-free Stv incorporates wt% amounts of $H_2O$; something that's not readily seen with Al-bearing Stv. The manuscript would benefit from discussing this a bit more. See, for example, Kueter et al 2023.

Thank you for your rigorous consideration suggestion. Many studies have already revealed that Stv in subducted slabs (open system) can contain a small amount of Al (up to 5 wt. %, Litasov et al., 2007) in its crystal structure because of coexisting Al-rich minerals such as garnet and its high-pressure phases. That's why we focus on water in Al-bearing Stv in our paper. However, we added discussion of wt.% amounts of $H_2O$ in Al-free Stv following your suggestion which improved the manuscript a lot.

Line 144 – "Statistically compiling…" I don't understand the wording. Figure 5 does not show any statistics or statistical breakdown of the data. It's a summary plot of available data. Also, please cite the

literature shown in Fig 5. I would recommend being more quantitative as well: Maybe you can provide a regression of the data and plot it in Fig 5 (similar to what Litasov et al. 2007 did).

Thanks. We clarify the statement of "Statistically compiling…". Also, we correct the cited literature and plot a regression line in it.

Line 167 onward - The fO2 part of section 4.2 is too vague and qualitative. There is no real information gain from it. I would recommend to discussing it in more detail, or (less recommended) omitting it.

We are very sorry for our negligence of this part. the $fO_2$ could have strong effect on the water solubility of some NAMs such as garnet (Zhang et al., 2022) containing valency elements (e.g., Fe) in the crystal structure. Since a small amount of Fe and Ti was determined in a pseudomorph of former stishovite from a continental subduction zone (Liu et al., 2018), we believe $fO_2$ could affect the water solubility of Stv. However, no relevant research has clarified this issue so far. Considering this is a review paper, we would like to give some insights for the research in the future even seldom been studied up to date. Anyway, we added some statements in more detail in the revised manuscript.

**Chapter 5 – Water solubility…**

The following is, in my opinion, the most important issue that has to be addressed:
This chapter is the core of the study and subsequent models rely on equation 2 given in it.
Consequently, Eqn 2 needs to be carefully derived and explained:

It is important to

- discuss the dismissal and use of literature experimental data in more detail

Thank you for your advice. We excluded the published water solubility in Al-bearing stishovite obtained by methods other than FTIR such as TGA/SIMS, and only included collected published FTIR data on $H_2O$ solubility contents in Al-bearing stishovite from FTIR measurements. We add this statement in the main text.

- show a plot that displays the data and the fits from which Eqn. 2 has been derived

Thanks. We have added experimental and computational data to Table S1 in the Supplement and Fig. A1 in the Appendix, showing a good correlation between them.

- make a thorough error/uncertainty analysis of the fit

Thank you very much, we added the uncertainty followed by one standard deviation for the obtained parameters of Eqn. 2.

- discuss the obtained fitting parameters (are they realistic, do they compare with literature?)

We gratefully appreciate for your suggestion. Many studies (e.g., Kohlstedt et al., 1996; Mierdel and Keppler, 2004; Karato, 2010) have described that the water solubility of NAMs (e.g., olivine, orthopyroxene, clinopyroxene) follows such a thermodynamic relationship: $C_{OH} = A \cdot f_{H_2O}^n \cdot exp\left(-\frac{\Delta E + \Delta V \cdot P}{R \cdot T}\right)$, where $C_{OH}$ is the water content, $A$ is a constant, $f_{H_2O}^n$ is the water fugacity, the value of n depends on the hydrogen incorporation mechanism, and $\Delta E$ and $\Delta V$ denote the energy and volume changes associated with hydrogen dissolution.

Following those basic thermodynamic rules, Zhao et al. (2004) summarized the relation of hydrogen

solubility of olivine as a function of temperature and iron content: $C_{OH}^{Ol} = A \cdot f_{H_2O}^1 \cdot \exp\left(-\frac{\Delta E + \Delta V \cdot P}{R \cdot T}\right) \cdot \exp\left(\frac{\alpha \cdot X_{Fa}}{R \cdot T}\right)$, where $C_{OH}^{Ol}$ is hydrogen solubility of olivine, $X_{Fa}$ is the mole fraction of fayalite, the other parameters have the same meaning as mentioned above.

Inspired by Zhao et al. (2004)'s work, we propose the relation for water solubility in Al-bearing stishovite as Eqn. 2 in the main text. Unfortunately, no similar model for stishovite has been proposed in previous studies. Therefore, we cannot make a comparison with literature. But theoretically, we believe this model is effective according to other NAMs (like olivine, orthopyroxene, and clinopyroxene) reported in previous works.

- discuss limits of the fitting method (e.g., boundaries, uncertainties)

Thanks very much. We used a non-linear least square fitting method to fit Eqn. 2. Mathematically, this method is widely used to solve the questions on relationship between a group of numbers. For water solubility in NAMs, this method is a common practice to fit the equation $C_{OH} = A \cdot f_{H_2O}^n \cdot exp\left(-\frac{\Delta E + \Delta V \cdot P}{R \cdot T}\right)$ in previous studies. Therefore, we believe the method we used to fit Eqn. 2 is convincing.

I further would recommend providing at least one example calculation (main text or appendix). This helps the reader tremendously to comprehend and reproduce the data & models. Also, please provide the exact literature data in the appendix that you used for the fitting.

Thank you, we added Table S2 in the Supplement showing the calculation for the cold subducted slab in Fig. 7. And we added the fitted literature data in Table S1.

Since the study focuses on Al-Stv only, it is ok to omit the high $H_2O$ experiments of Al-free Stv, but it would be good to mention this in the text and give reasons for it.

Thanks, we added more discussion on the high $H_2O$ of Al-free Stv in the main text.

Last but not least: Eqn 2 has a typo ("epx", same in the conclusion part)

We are very sorry for our incorrect writing in this manuscript. We checked and corrected the typos in the main text.

**Chapter 6**

The second half of the first paragraph should also cite Walter (2021, Natl Sci Rev, 2021, Vol. 8).

Thank you for your suggestion. We cited Walter (2021, Natl Sci Rev, 2021, Vol. 8) in the Line 264.

Fig. 6 - I am not really sure what Fig. 6a is based on. Also, range is 800-1400 °C, caption says 1000-1400 °C). 6b – the curve maxima are basically the Stv to CaCl2-Stv phase transitions? Please explain in more detail. Maybe mark the transition interval in the diagram.

Thank you very much. Actually, Fig. 6a and 6b illustrate the calculated water solubility of Stv at different temperature and pressure based on Eqn. 2. The range of temperature shown in original Fig. 6a is 800-1400 °C (1000-1400 °C in the caption is a typo). However, we added the calculated result at 1600 °C to

both Fig. 6a and Fig. 6b in the revised text. The curve maxima in Fig. 6b shows the water solubility of stishovite at conditions close to MTZ. We are not sure if this corresponds to the phase transition from Stv to $CaCl_2$-Stv. Since the pressure at where the Stv transition to $CaCl_2$-Stv remains debated (Fischer et al. (2018) suggested 68-78 GPa, Ishii et al. (2022) suggested ~24 GPa), we are not trying to mark the phase transition line in Fig. 6b.

L 224 – Please refer to fig. 7.

Thanks. We added.

L 225 – "Consequently, Stishovite…" The statement in this sentence is correct, but not very new. Several previous studies concluded this, e.g., Walter 2021 and refs within. That should be cited accordingly.

Thanks for your suggestion. We cited the previous studies (e.g., Lin et al., 2020; Nisr et al., 2020; Walter, 2021).

L 228 – "Taking into …" Please elaborate on the melts and outline your reasoning a bit more in detail.

Thank you for your suggestion. Hydrogen as incompatible element flavors melts more than minerals. Previous studies suggest hydration of a large region of the transition zone and that dehydration melting may act to trap $H_2O$ in the transition zone (e.g., Schmandt et al., 2014). Therefore, the melt in the mantle transition zone could reasonably unload the water carried by Stv. We add more statements in the main text.

Figure 7 – "Yin and Kang, 2023 (in preparation). I disagree with citing non-peer reviewed work. Is there an alternative?

Thank you for pointing out this issue. We replaced this citation with Lu and Keppler (1997) who contributed a model for water content in garnet.

**Paper mentioned:**

[revised manuscript text omitted]

---

## Author Comment (AC2)

The work reviews the hydrogen solubility in $SiO_2$ stishovite and the implication behind it. I like to read it and I think the authors took in consideration all the relevant literature to write down such review. I also liked that they also proposed a new possible equation for calculating the solubility of hydrogen in stishovite. The work is written very well and the science is sound. So I recommend to publish it with only very minor corrections, which are reported here below.

Line 34-36. Well, here I would definitively cite Gu et al. (2022) Nature Geoscience, who showed that coesite (former stishovite) was present within a natural super-deep diamond formed at the boundary between the transition zone and the lower mantle.
Thank you very much for your suggestion. We cited Gu et al. (2022) in the revised text.

Line 51. Please use the accepted nomenclature for space groups (so the Bravais lattice, and the glade and mirror planes must be reported in Italic)
Thank you for pointing out this problem in manuscript. We corrected it.

Line 215. "Gavrilenko (2008)" instead of "Gavrilenko, 2008 showed…"
Thank you so much for your careful check. We corrected it.

I have doubled checked the references and I only found the following minor issues:
- Bolfan-Casanova et al., 2000 (in Table 1) is mentioned in the text but is missing in the reference list
Thank you very much. We added Bolfan-Casanova et al. (2000) into the references list.

- Stishov and Popova (1961) is not quoted in the reference list
We are very sorry for our negligence. We added it.

- Yin and Kang, 2023 check with the journal if this can be reported
Thank you very much. we replaced this citation with Lu and Keppler (1997) who contributed a thermodynamic model for water solubility in garnet.

**Paper mentioned:**

Bolfan-Casanova, N., Keppler, H., and Rubie, D. C.: Water partitioning between nominally anhydrous minerals in the $MgO–SiO_2–H_2O$ system up to 24 GPa: implications for the distribution of water in the Earth's mantle, Earth and Planetary Science Letters, 182, 209–221, https://doi.org/10.1016/S0012-821X(00)00244-2, 2000.

Gu, T., Pamato, M. G., Novella, D., Alvaro, M., Fournelle, J., Brenker, F. E., Wang, W., and Nestola, F.: Hydrous peridotitic fragments of Earth's mantle 660 km discontinuity sampled by a diamond, Nat. Geosci., 15, 950–954, https://doi.org/10.1038/s41561-022-01024-y, 2022.

Lu, R. and Keppler, H.: Water solubility in pyrope to 100 kbar, Contributions to Mineralogy and Petrology, 129, 35–42, https://doi.org/10.1007/s004100050321, 1997.

Stishov, S. M. and Popova, S. V.: A new dense modification of silica, Geokhimiya, 10, 923–926, 1961.

---

## Author Comment (AC3)

This manuscript delivers a comprehensive review of the controlling factors and incorporation mechanism influencing water solubility in stishovite. Furthermore, building on published findings, it introduces a novel equation for estimating water solubility in Al-bearing stishovite. The manuscript is generally well-structured, and the presented results are robust. The discussion regarding the role of water in stishovite contributing to the presence of a water-rich transition zone appears sound. Therefore, I recommend its publication in Solid Earth after some revisions.

I have only the following minor concerns and suggestions on the presentation of results and interpretations.

Page 2, line 47. "Under average geothermal gradient". The average geothermal gradient mentioned here is often associated with specific geological settings. I assume the authors are referring to the temperature profile along the surface of oceanic subducting slab. It needs clarification, and the different types of subduction zones (cold subduction, hot subduction) should be labeled on Figure 3. Another simpler approach could be to mark the geotherms of 5 degrees Celsius/km, 10 degrees Celsius/km, and 15 degrees Celsius/km, respectively, on Figure 3.

Thanks very much for your suggestion. We plotted the subduction slab geotherms in Fig. 3 according to Zheng et al. (2016).

Page 3. Figure 2. The data source for the red geotherm curve needs to be indicated.

Thank you for pointing out this issue. The data source for the red geotherm curve is from Kaminsky (2012). We added the reference.

Page 3. Line 64. "only found". It may not be accurate. Yang J.S. et al. (2007, Geology, doi: 10.1130/G23766A.1) identified polycrystalline coesite as a potential pseudomorphic replacement of stishovite in Tibetan chromitites. Therefore, it is preferable to omit the word "only."

Thank you very much We removed the "only" and added Yang et al. (2007) following your suggestion.

Page 10. Figure 6. Here, the mantle transition zone is only associated with pressure. In reality, the temperature in the mantle transition zone may be around 1400-1600 degrees Celsius (Ito &Katsura, GRL, 1989), exceeding the temperature range calculated in the figure. If you want to correspond to the temperature and pressure of the subducting slabs in the mantle transition zone depths, it would be necessary to separately label the temperature and pressure ranges for different types (cold and hot) of subducting slabs.

We gratefully appreciate for your valuable suggestion. The temperature range in Fig. 6 is extended to 800-1600 ℃ for the calculation through Eq.2 in the revised text. As shown in Fig. 6, the water solubility in Al-bearing stishovite is slightly affected by temperature but strongly controlled by pressure, exhibiting a positive correlation below 22-32 GPa and a negative correlation above 22-32 GPa (Fig. 6b). No matter what types (cold and hot) the subducting slab is, a maximum water solubility is reached at the conditions around the bottom of mantle transition zone according to our calculation. Therefore, we just plotted the pressure and temperature dependence of water solubility in Al-bearing stishovite.

Page 10. Line 224. "along the geotherm". The term "geotherm" here needs clarification to specify what exactly is meant by "geotherm" (mantle geotherm or subduction zone geotherm).

Thank you for your comments. We clarified the geotherm as subduction slab geotherm in the revised text.

Page 11. Figure 7. In Zheng et al. (2016), four distinct geotherms (ultracold subduction, cold subduction, warm subduction, and hot subduction) are identified. It is essential to specify which geotherm was employed and provide a rationale for the choice. Considering the comprehensive coverage, it might be more beneficial to utilize the full range of geotherms to encompass variations.

Thank you for your rigorous consideration suggestion. The geothermal gradient we selected for the calculation in Fig. 7 is a cold subduction gradient from Litasov and Ohtani (2007). Moreover, the calculation result base on the hot subduction gradient from Litasov and Ohtani (2007) is shown in Table S2 in the Supplement and Fig. B1 in the Appendix.

The language expression in this paper is overall very good, but there are still some places that may need modification. I have made some annotated suggestions in the attached PDF file for the authors' reference.

Thank you very much. We merged your modification in the revised text and further polished the language in the revised text.

**Paper mentioned:**

Ito, E. and Katsura, T.: A temperature profile of the mantle transition zone, Geophysical Research Letters, 16, 425–428, https://doi.org/10.1029/GL016i005p00425, 1989.

Kaminsky, F.: Mineralogy of the lower mantle: A review of 'super-deep' mineral inclusions in diamond, Earth-Science Reviews, 110, 127–147, https://doi.org/10.1016/j.earscirev.2011.10.005, 2012.

Litasov, K. D. and Ohtani, E.: Effect of water on the phase relations in Earth's mantle and deep water cycle, in: Advances in High-Pressure Mineralogy, Geological Society of America, https://doi.org/10.1130/2007.2421(08), 2007.

Yang, J. S., Dobrzhinetskaya, L., Bai, W. J., Fang, Q. S., Robinson, P. T., Zhang, J., and Green, H. W.: Diamond- and coesite-bearing chromitites from the Luobusa ophiolite, Tibet, Geol, 35, 875, https://doi.org/10.1130/G23766A.1, 2007.

Zheng, Y., Chen, R., Xu, Z., and Zhang, S.: The transport of water in subduction zones, Sci. China Earth Sci., 59, 651–682, https://doi.org/10.1007/s11430-015-5258-4, 2016.